# Navigating the Impossibility of Universally Ethical AI - a Practical Guide for Researchers

Anonymous Full Paper
Submission

## Abstract

The area of ethics is full of trade-offs, but this is sometimes ignored in Machine Learning research. I argue that there can not be a single way to create ethical AI, agreed upon by humanity at large, but that this can be handled by considering a broader range of stakeholders and clearly stating assumptions, values, and trade-offs. I begin this paper with a survey of different current approaches to AI ethics and some of their drawbacks. I go on to highlight the impossibility of a universally ethical AI based on the inherent contradictions in plural human values and directly contradicting definitions, as well as to provide some discussion on the authoritarian connotations of aims to find a singular true morality. Then, some approaches to handling this impossibility through a democratisation of the problem and clear communication are proposed. Lastly, a practical guide to navigating moral issues in machine learning research – based on dialogue, transparency, and conscious trade-offs – is proposed in the form of four basic principles and a checklist.

## 1   Introduction

Ethical issues in the development and deployment of AI tools have been increasingly researched in recent years, with specialised journals [1, 2], dedicated conferences [3–5], and increasing work on national and international guidelines for ethical (or trustworthy) AI [6–8]. The increased interest is a positive sign that the issues are taken seriously, but most works are either heavily theoretical, suggests a singular solution, or are clearly aimed at the implementation and large scale developments at companies (e.g. the EU guidelines [6]). By this, there tend to be a gap between the philosophy of AI ethics, and the implementation of it [9]. The aim of this paper is not to solve the issue, but to begin bridging the gap by presenting a variety of approaches from philosophy to a more technical audience, together with guidance on how these can be considered in ML research.

A complication in applying methods for ethical AI, is that it is impossible to come to a universal agreement on what constitutes ethical AI. The multitude of ethical theories and definitions of values, and the complexity of real world issues, makes trade-offs both necessary and non-trivial. This is an issue present in most ethical considerations, but in an AI setting the myth of neutral tech can easily mask the issue, if one is not careful. This impossibility, and the necessity to acknowledge it, is my first thesis of the paper.

However, acknowledging the impossibility of universally ethical AI is not enough in and of itself, although it is an important piece in accomplishing responsible AI practices. Even if no single approach will be perfect by anyone's standard, there are ways to make better decisions. Or rather, to let others make better decisions. I argue that the responsible researcher need to consider a multitude of different views on the ethical issues in their research field, and take responsibility for and clearly communicate the prioritisations and trade-offs decided upon. This is my second thesis of the paper.

The paper begins by a survey of a variety of established approaches AI ethics. These are presented together with some examples on how they can be used in AI development, and some motivations and common critiques. Thereafter, I argue for the impossibility of finding a singular solution: the first thesis.

Further, I present some democracy based approaches to AI ethics, and two frameworks for analysis: ACROCPoLis and the EU trustworthy AI assessment list. Based on these, I argue that an important element of AI ethics is to make conscious trade-offs and communicate these clearly to allow for wider discussions both within the AI community and among the larger public affected by AI implementation.This is my second thesis of the paper. This is followed by a practical guide for AI researchers, with some basic principles and a checklist based on the previous arguments of the paper. I conclude by a summarising discussion.

## 2   Two main approaches and their critiques

Many works on ethical AI rely on a utilitarian approach of maximising a utility, or minimising a cost [10], and including different aspects and stakeholders in this formulation [11, 12]. Another prevalent approach is to formulate principles for AI developers to lean against, such as justice, privacy, and explainability [6, 7]. However, according to works

of moral philosophy and philosophy of AI, there are several issues with these approaches, making them unsuitable as universal solutions.

## 2.1 Utilitarianism, unfairness, and immeasurability

Utilitarianism is one of the most famous and well-accepted, ethical frameworks in the field of Ethics. It is a version of consequentialism, where the consequences are the only thing determining the righteousness of an action, with the specific goal of maximising overall utility [10]. Although utilitarianism is a well-established framework, nothing is preventing the "utility" from being unfairly distributed. A well-known example is the Utilitarian monster [13], a thought experiment where one individual experiences significantly more pleasure from resources than anyone else. There is no single person feeling the same amount of pleasure from a piece of food as this utilitarian monster, no person appreciating a cent or a loving word as much. In a utilitarian decision, any ethical action have to be providing for his monster, since this will always maximise the total pleasure of the word.

Another critique can be found in the novella *Those who walk away from Omelas* by Ursula K. Le Guin [14]. Here we get introduced to the perfect society of Omelas in the middle of a festival. Everyone is happy and content. There are no wars nor soldiers, but plenty of food and love. If you can think of anything to make it better, more pleasurable, it is already there. Not a single person feels pain - or no, a single child feels pain. They are locked in a cellar with just enough food to eat, without any human interaction and living in their own excrement. They are miserable, but have given up screaming for help. And they are the guarantee for the happiness for the rest, if their misery where to be relieved the perfect society would fall. So people accept the necessity and let them be. A utilitarian approach.

Another problem with the utilitarian approach lies in deciding whose utility counts. The pleasure of humanity is probably the standard utilitarianism answer, but it is not unproblematic. Varying definitions on who counts as part of humanity have been used to discriminate people for a large part of our history, which might make this an unsuitable term. Even without these connotations, there are other complicating aspects.

In a machine learning algorithm, it is often easiest to only include people directly affected by the algorithm; the people whose utility can not be changed by the system do not need to be considered. Nonetheless, there are people affected indirectly that are easily forgotten. For example, ignoring the working conditions of miners and data-labellers and the environmental impact of large-scale computing makes the development easier.

Further, the utility of each actor is not necessarily weighted equally. This can be a deliberate choice, but is often an effect of the accuracy of predicted outcomes being biased due to data access and historical discrimination[15]. There is a lot of research currently going into the latter problem [16], but the truth is that there will always be a trade-off between fairness and accuracy on biased training data [15]. It is not obvious how these should be handled properly [15, 17].

Lastly, putting numbers on pleasure is not a simple task. In healthcare, measures such as Quality Adjusted Life Year are used to make decisions about treatment prioritisation but have also received a fair amount of critique, for example, due to treating the life of people with disabilities as less valuable when the years are quality-adjusted [18]. Similarly, it might be hard to put comparable numbers on private data and discovering new knowledge.

Further, in many commercial machine-learning settings the utility-measures chosen and trade-offs done are hidden in large code bases and their intricacies treated as trade secrets, discouraging public discourse and promoting the idea of neutral tech. When researchers don't present their decisions on ethical issues during development, similar issues arise.

## 2.2 Guidelines, minimum efforts, and loopholes

The approach of guidelines has its own problems. They are often hard, or even impossible to implement in practice, and enable minimal-effort solutions.

Due to vague formulations, there is often no specified minimum requirements for the different values. The concept of privacy is a common value, lifted in both the EU and UNESCO guidelines [6, 7], however the level of privacy is not explicitly stated. Maximising privacy would mean to never use any personal data in AI algorithms, but this is not necessarily the goal. There may also be both minimal legal requirements and ethical principles to follow about the same value. For example, the EU have clear laws, the GDPR [19], on how private information can be used, but still include "Privacy and data governance" as a principle under ethical AI in the guidelines for trustworthy AI [6], not only as part of the legality of the AI. How these values are interpreted also affect research directions; currently researchers develop ways to share anonymous data between data owners through methods such as federated learning [20], and differential privacy [21], which has a clearer connection to GDPR rules on sharing data with third parties, than to existential questions on how sharing personal data with *any* data owner affects ones sense of integrity (for the

latter view, see e.g. [22]).

Another limitation on the concept of guidelines is found when asking "who writes them, and for whom?". Both the EU and UNESCO guidelines are written by teams of AI experts, which is highlighted as a strength. In some ways it is, but it is also important to consider which voices may have been left out when asking highly educated people in a technical field to make the ethical considerations. Secondly, many guidelines seem directed mainly towards companies with large scale projects. As research plays an important role in the development of new methods, one might consider the lack of focus on more basic research in the area to be an issue.

Finally, the successful usage of guidelines requires the actors to actually aim at being fair and ethical, since otherwise it can lead to fairwashing; the concept of making a model seem fair or ethical when it is not. A clear example of this is LaundryML [23], developed to give a fair explanation of any decisions from a black box algorithm, even when the original decision is made on biased data, e.g. salary decisions based on gender being explained with reference to factors such as higher education and marital status. It is probable that other similar loopholes exist.

# 3 Alternative approaches

Although the utilitarian and principle-based approaches to AI ethics are common, there are works lifting other ethical considerations in this area. These are not necessarily theories about making the right decisions, but rather frameworks for taking other aspects into account in the ethical analysis.

## 3.1 Power, relations, and care

Lacking in utilitarian and guideline based approaches is the study of power and relationships. One could argue that these are not relevant, but just as well that they are. Take the example of the child in Omelas again. A fully utilitarian approach would suggest this is okay since it maximises overall happiness, and that the situation would be the same if the suffering was not done by a powerless kid, but by a leader who took the suffering upon themself to further the prosperity of the people. However, the power dynamics in these two situations are completely different, and many would argue that suffering pain to help overall happiness is strictly more moral than forcing the same amount of pain upon someone else for the same end. In the AI community, there are several relevant power dynamics between companies, users, data labellers, researchers, mineral miners, and governments, that can affect what we deem ethical. The issues of power dynamics and interpersonal relationships have been thoroughly discussed in various branches of feminist ethics; a rich subfield of ethics philosophy.

One theory more comparable to utilitarianism is care ethics, generally seen as a type of feminist framework, which says that ethical decisions should be based on care and empathy. There are a few papers in which this is proposed as an alternative framework for AI ethics. In [24], care ethics is proposed as a way to handle the moral distance that automated decisions may lead to. When decisions about people are made with little human contact, care has to be explicitly encoded in the moral judgements as they will not naturally arise from the human connections (as they are not present). By incorporating the notion of care and considering the context of each decision, they argue that more stakeholders will be taken serious in the implementation process. In [25] ethics of care is viewed as the opposite to oppressive AI development strategies based on capitalism, and power seeking agents.Care is also argued to be important to achieve explainable and transparent AI accessible for more diverse groups [26], and to understand existential aspects of incorporating AI in day to day life [22].

## 3.2 Climate, resources, and nature

Environmental ethics is a broad field of research, but the common theme is that nature should be considered in moral judgements. In shallow environmental ethics the value of the environment comes mainly from in which ways it can help human societies to flourish, while the subfield of deep environmental ethics arises from the inherent value of the environment itself and anything living in it [27]. Similar to feminist ethics, environmental ethics is more about what issues to highlight and value, than about exactly how to make the final decisions.

As large-scale computing requires a lot of power [28] and many natural resources [29], environmental ethics can be directly applied by considering these aspects in more depth when considering the ethicality of ML models. There are several works arguing for more significant incorporation of these values, either by calls for environmental ethicists to take a larger interest in AI [30], by highlighting trade-offs between ecological and human centred values [31], and by encouraging clearer measurements of the climate impact of model training [28].

## 3.3 Incorporation of non-WEIRD values

The acronym WEIRD stands for Western, Educated, Industrialised, Rich, Democratic, and is used to describe cultures fulfilling these criteria. Current work on aligning AI with human values have received critique for only focusing on WEIRD values, even

though these are in a minority (based on population size). For example, it has been shown that ChatGPT have both values and ways of thinking significantly more closely related to countries such as the United States and Sweden, than to less WEIRD countries such as Ethiopia and Libya [32].

There have also been works aiming to incorporate non-WEIRD values into the discourse. The Ubuntu philosophy has been suggested as a source of inspiration, with its clear focus on helping the group and valuing its social relationships [33]. The concept of *honour* is common in many different non-WEIRD societies, and has been lifted as a way to view AI ethics in a more global lens. This could be done through considering the preservation of honour, and viewing content moderation more in terms of a guardian protecting the users, rather than a form of censorship, regardless of which values are prioritised [34]. However, grouping such diverse values into a single category of non-WEIRD ethics does have its own problems.

## 4 The inherent contradictions

As shown in the previous sections, there are many ways to conceptualise ethical AI, each with its own critiques. The critiques for specific views is not the only complication of a universally ethical AI though. In this section I argue that a universal approach is inherently unethical; my first thesis.

### 4.1 Trade-offs and contradicting values

In AI ethics it is common to pick some sort of values to be considered. In the approach of guidelines they are often stated as explicit values, but in a utilitarian approach one can see from Section 2 that decisions have to be made on how things are valued, in terms of how things can be counted as utility. When several things to value are picked however, they may contradict each other. One example is the common values of privacy and explainability, present together in for example the EU guidelines on trustworthy AI [6]. It has been shown that some methods for generating explanations risks leaking information about the training data and thus violating privacy [35]. This may be a more general problem, as a valid explanation require a basis in the training data.

### 4.2 Contradicting definitions

Even if all humans could agree to some set of values, implementation is not straight forward, due to inherent contradictions. As discussed in the previous section, different values can contradict each other, but even a single value have room for contradictions, based on the vast number of ways in which

they can be defined. A simple case is values that act as constraints, but where the cut-off may be different depending on who is asked. For example, some could say that 100% accuracy on test data is a sign of a too small test data set, or that the model risks being sensitive to slight changes in the data generating process, while someone else might require 100% accuracy to deem the model trustworthy.

A slightly more complex example is that of the fairness paradox. Three well established definitions in algorithmic fairness are equalized odds (false positive/negative rates do not depend on protected characteristics), predictive parity (the predictive power do not depend on protected characteristics), and counterfactual fairness (the model should have no causal effect between protected characteristics and outcome), and it has been proven that these can not occur at the same time [36]. Even definitions that contradict each other can be gradually improved on, but this again requires making trade-offs between them [37].

### 4.3 The power to decide

From Sections 2 and 3, it is clear that human values are not universal, and that there is a multitude of ethical frameworks. This means, inherently, that there is no superior ethical theory. Assuming that there is would ignore millennia of ethical debates and the diversity of human cultures. There have certainly been efforts to define a singular ethics and a universally correct way of living, but historically many such attempts have supported colonialist efforts and become an excuse to treat people with other ways of living as morally inferior.

This is also a clear problem in current efforts of globally used AI models with "human-aligned" values, which are not representative of all parts of humanity [32]. This is also a clear problem when a select number of experts try to figure out AI ethics among themselves, by developing guidelines or writing papers for exclusive conferences. However many AI experts and ethics professors put their heads together, most people are still not a part of the discussions, and do not have a say in the process. In the AI community, we can support any number of values such as "equity", "reducing bias", and "privacy", but without talking to different stakeholders and considering a pluralistic view on ethics, it will still be an ethics for the few. For the groups represented among the experts, and the people we can imagine.

A related question is what happens when a small number of pre trained AI systems is implemented in a large set of tasks, similarly to what we can see with Large Language Models (LLMs) today. Even assuming they represent some sort of average or generally agreed upon ethics, small everyday decisions are suddenly taken using a universally adapted ethical

system, instead of a multitude of more intuitions.

## 4.4 A future: The universal ethical validation layer

Consider a potential future, a few years from now. The field of AI ethics has continued to grow, but the conclusions in the large journals begin to coalesce on a set of clearly defined values. A group of Machine learning researchers see their chance to make an actual improvement in the world, by creating the *Universal Ethics Assessment Layer*.

They review the top cited papers on ethical AI and create a layer that can create an ethicality score for any machine learning model output. The score is based on formalised concepts of values such as fairness, honesty, and justice, and the importance of them weighted according to their prevalence in the surveyed papers. The layer is made to be applied to deep learning architectures, taking in the input and the output of the main model, as well as the results of querying the model for different inputs for comparison. Based on this an ethicality score is provided, which during training is fed back to the model as a reward until the ethicality score is consistently over a certain threshold. If the threshold is violated during deployment, the model will randomly make small, temporary tweaks to its weights until an output with a valid ethicality score is achieved.

The results are presented in a paper in Nature, with validation of the model on a large variety of architectures and tasks, and showing accomplishment of a higher ethicality score than humans on comparative tasks, from image generation and chatting, to cancer diagnosis and mortgage approval. The paper is soon one of the most cited in the field of AI ethics, and newspaper headlines read "The alignment problem is solved". The work receives some critique for setting fixed weights on the different values, but the debate dies quietly with the developers answer that the weights can be easily changed in their open-source code. However, no technical user seems to bother.

As a part of efforts for more ethical AI research, many of the large ML venues start requiring usage of the Ethical layer in submitted work. On the website of one of the largest conferences on machine learning, one can read

"To make accuracy stay a useful tool in the evaluation of machine learning methodology, it is important that ethicality can not be traded for accuracy. Therefore, each submission aiming to improve accuracy or developing new architectures must include the ethical layer in their model, with the original weights and an ethicality threshold of 15. Any non-motivated deviations from this will lead to desk rejection."

The ethical layer further becomes a simple way for companies to show compliance with legal requirements on trustworthy and responsible AI, and soon this layer is a basic building block in all machine learning systems. The same ethical basis is used all over, and when a healthcare provider is accused of racism they simply point to the ethicality score of the treatment decision, and the issue is settled.

Since models are now verifiably ethical, people start to imitate them in their personal moral decisions. As time goes and the universal ethicality is more widely applied, people forget how essentially, a few AI researcher single-handedly defined ethicality all over the world.

This may in some senses be an appealing solution, but it also takes away something of the plurality of human views. When the same ethical trade-offs is propagated through all parts of society and AI tools are harder to avoid, minority opinions become less reflected and marginalisation risks increasing.

# 5 Democratic approaches to AI ethics

One way to handle these impossibilities is to take a more political view of the issues, and consider democratic approaches to AI ethics.

## 5.1 Democratic regulation

One straight forward way to democratise AI is by democratically constructing leagal requirements for its development, research, and deployment. Uncommon opinions, minority groups, and people not residing in the country may still be missed, but by voting a larger part of the population has a say. One example of legal regulation is the EU AI Act, which regulates how AI systems can be used, together with parts of GDPR when personal data is concerned. Arguments for regulation via legal means have been lifted before, both based on the concept of ethicality as inherently political [38] and the idea that self-regulation from tech companies centralises the power over AI systems even more [39].

## 5.2 Transparency

A core part of a democratic society is transparency that allows the public to question and analyse political decisions, and the actions of others in power. When AI systems are used in more sensitive situations and given more autonomy, the transparency of them becomes important by the same reason. Note that this transparency is not equivalent to common notions of explainability, which often focuses on explaining specific decisions. Rather, the democratic transparency requires public access to information about what values are encoded in the machine, and

how. Visiting the website for *chatGPT*, the information provided about value alignment is significantly lacking. This is a part of their explanation:

> We randomly selected several alternative completions, and had AI trainers rank them. Using these reward models, we can fine-tune the model using Proximal Policy Optimization.

and another states that

> We're using the Moderation API to warn or block certain types of unsafe content, ...

where the goal of the mentioned Moderation API is to flag content that is sexual, hateful, violent, or promotes self-harm. Nowhere do they give details on who the AI trainers were or what values they prioritised, nor what they define as hateful and why they decided to flag exactly those four as unsafe content.

To be able to have thorough discussions on the values encoded in models like *chatGPT*, they need to be more transparent than this. If the data trainers where all male college students in the US, quite many human perspectives are missed in the fine tuning process. Exactly what data was used to train the original model is also not publicly available.

### 5.3 Participatory design

Another approach to involve more people in AI development is through participatory design. The concept is wide and can mean anything from surveying a few colleagues about an interface design choice, to letting a representative (by some definition) group fully own the project. A recent survey of participatory design in AI development noted that research project often do not enable the extra time needed to fully involve a part of the public, which has led to smaller scale involvements or using proxies, such as a stand in person for another group or a ML model trained on different peoples preferences [**dem:delgado2023participatory**].

## 6 Frameworks for analysis

Several different frameworks have been proposed to analyse the ethical implications of AI models. Here I present two; the assessment list from the EU expert panel and ACROCPoLis from a recent scientific paper.

### 6.1 Trustworthy AI assessment list

As a part of the EU guidelines for trustworthy AI, an assessment list was released [6]. The list is several pages long and consists of concrete questions to ask when developing an AI system, such as: "Did you assess to what extent the decisions and hence the outcome made by the AI system can be understood?", "Did you establish mechanisms to ensure fairness in your AI systems? Did you consider other potential mechanisms?", and "Did you assess whether there could be persons or groups who might be disproportionately affected by negative implications?".

The idea is that when developing or implementing an AI system the list should be used to verify that all aspects of the guidelines have been considered. The list is thorough and captures all the different parts of the guidelines, but are at points vague, clearly focused towards larger corporations, or without specified results, e.g. questions of the form "Did you assess...".

### 6.2 ACROCPoLis

ACROCPoLis is a recently developed framework to assess the ethical implications of an AI system [40]. For good and bad, it is significantly broader in formulation than the above mentioned assessment list. The idea is to identify the categories Actors, Context, Resources, Outcome, Criteria, Power, and the Links between them. This is in many ways based on a feminist ethics with its focus on context and relations, but it also aims to facilitate viewing issues from different perspectives by singling out the differences in context and power between the stakeholders (actors). However, it does not provide any clear answers about ethicality, but rather a language to discuss it.

## 7 A practical guide

To conclude, I provide a practical guide to navigating the complex questions of AI ethics in everyday research activity. A few basic principles are derived from the arguments this far, which is then implemented as a short checklist inspired by the ACROCPoLis framework and the EU assessment list. Finally the checklist is applied to a hypothetical research project.

### 7.1 Basic principles

Based on the discussion this far, I propose four basic principles for responsible AI research.

**There is no universal morality.** As argued in the previous section, a universal ethics for AI is inherently impossible. To still perform responsible research in the field of machine learning, this needs to be both accepted and acknowledged. Assuming that there is a singular solution to ethical AI risks authoritarian consequences and further marginalisation of minority groups.

**Each stakeholder deserves consideration.** This is a major part of the democratic view on ethics,

---

## Responsible ML research

### Stage 1: Identify potential impact

☐ Consider the potential applications of your research project, and how likely they are. Can you stand behind them with good conscience?

☐ Who would deploy or use this type of system, and what are the power dynamics?

☐ What technologies can this knowledge enable, and who does it affect?

☐ Where does your data come from, are there data labellers and/or unknowing subjects involved?

☐ What resources do you use, what is the climate impact, and who does this affect?

### Stage 2: Gather different perspectives

☐ See if you can get a chance to talk to the identified stakeholders through personal contacts, public events, or online communities.

☐ Otherwise, read works by them or journalistic reports, and imagine yourself in their place.

### Stage 3: Make transparent trade-offs

☐ Make trade-offs based on your ethical values, but remember the position of power this puts you in.

☐ Clearly present these trade-offs in all communication about your research, whether it is in a scientific paper or public outreach.

### Stage 4: Community engagement

☐ Ask other researchers about their ethical stance and the trade-offs of their research at conferences.

☐ Discuss ethical issues with colleagues.

☐ Engage in public dialogue and help unveil hidden values of discussed ML systems.

**Figure 1.** The checklist

---

but is also necessary for both determining utility distributions and to follow the feminist approach. One part of this is to work towards identifying all relevant actors; not only those directly affected by an algorithm, but those affected by the data acquisition or resource usage. Animals and the earth itself can also be included here, to use tools from environmental ethics.

**There will be trade-offs.** Most technology, and most actions, have a wide array of consequences, and often some are wanted and others unwanted. This means that trade-offs are necessary. There is no strictly correct way to choose these, but by communicating the choices they can be questioned, criticised, or reproduced. Ethical choices are hard, but unavoidable. Choosing to step away from a research topic is also a moral judgement.

**Silence is a political decision.** It is not the job of a researcher to decide what is the right usage of technology, but as the ones at the forefront of the knowledge researchers are needed to explain and point out ethical dilemmas and potential consequences. Especially, decisions on ethical judgements in AI development and research need to be communicated, otherwise it is close to impossible for others to consider or criticise the view. To not communicate these things, or not take part in public discussion, is a political choice, especially considering the importance of knowledge and journalistic investigation in a well functioning democracy.

### 7.2  The checklist

In Figure 1, the basic principles are combined into a checklist to be used in everyday research activity. The format is inspired by the AI assessment list, but with more focus on actions, fewer points, and a clearer focus towards smaller scale research. The ACROCPoLis framework is incorporated with actions corresponding to identify the contextual and power dynamics information of the problem, with an extra focus on identification of relevant stakeholders as both values like privacy and utilitarian approaches also require knowledge of the actors in the system. The aim of the checklist is not to favour specific ethical values, apart from the basic principles from the previous subsection, but rather to guide the researcher to make their own, informed decisions based on their ethical values, while still emphasising how this puts us in a position of power.

The checklist is divided into four stages, mainly based on when in the research project they would come into play. In the first stage, potential stakeholders and impact on them should be identified. This is not as action focused as the later stages, but is necessary to do beforehand. The questions are mainly aimed at enforcing consideration of different aspects and groups. The second stage is about gath-

ering the perspectives of the actors identified in the first stage. This allows for a deeper basis of understanding to base the later trade-offs on. The third stage is where the researchers own ethical stance comes into play. As has been shown there is no universally correct approach to ethical AI, so here the researcher must use their knowledge from the previous stages together with their own values to decide what trade-offs to make. However, to enable an ongoing debate on the ethical values in machine learning, all of these trade-offs need to be thoroughly documented and communicated. The forth stage is a bit outside of any specific research project, but rather a continuous work to foster a research culture in ML where ethical trade-offs are made open to scrutiny and debate among both researchers and the broader public. In the Appendix, an example is provided where the proposed framework is applied to a hypothetical research project by a fictional research group. It is omitted from the main paper for consistency, but may provide a more digestible interpretation of the checklist.

## 8 Discussion and conclusion

Ethics is a complex subject without simple answers, in AI just as in any other branch of human experiences. In this paper I presented a variety of approaches to ethics and how they can, and have been, applied to the field of AI and machine learning. Contradictions arise both from the plurality of the approaches and disagreements about definitions of fundamental values. What makes the field of ethics for AI somewhat unique is that many ethical decisions are implicitly made by researchers developing new algorithms, potentially without realising it. A simple assumption that notions of fairness can be directly incorporated into a utility function is an indirect utilitarian view on the issue, while silence on environmental costs for network training is a decision on which actors matter. Being a researcher in this field is also being in a position of power, with agency to shape future research directions as well as the public understanding of the algorithms we implement. I propose that we openly acknowledge the inherent impossibilities of singular technical solutions to ethical AI, and instead take responsibility for the trade-offs and values we decide to base our work on. Further, I suggest we give away some of our implicit power by communicating more transparently on ethics issues and lifting the voices of groups less often represented in the AI community.

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

# A  Application of the checklist to a hypothetical research project

**Disclaimer:**  *This is a hypothetical research group and research project, meant to illustrate how the checklist can be used. The concerns arising is not based on any actual persons opinion, and the actions taken are not necessarily the best choices by any definition.*

The research group $X$ has an idea for a new method to detect anomalies in sparse sequential data. They hope this can be used to improve automated health care suggestions for patients that take regular tests during pregnancy, cancer treatment, or type 2 diabetes controls. Before implementing anything, they go to stage 1 of the checklist.

**Stage 1:**  When considering potential applications, they realise that these can be very broad; from detection of credit card fraud to extraction of personal information. The first would be a relatively easy change of application, while the latter would require significant additional efforts. With the healthcare application they initially had in mind, they think hospitals would be the main users, or more probable companies selling the tools to the hospitals. Here there are power dynamics both between the companies and hospital, the hospital steering committee and the doctors, and between doctors and patients, with the patients in many senses furthest down the hierarchy. The data would come from a nearby hospital that the group is collaborating with, where data comes from patients and is labelled by doctors and nurses. The plan is to use anonymised data so that they do not need to ask for patients permission and go through ethics reviews. The model training do not require excessive compute power.

**Stage 2:**  The group schedules a meeting with some doctors and nurses at the hospital, all related to type 2 diabetes care. Some of the doctors are very positive to decision tools to help them make correct decisions under time pressure, while some of the nurses are afraid the patients will feel less comfortable when their data is fed to a machine, even if it stays locally. The doctors that have had collaborations with the research group before is mostly excited by the potential of revealing new patterns in the data they already use for decisions. Patients are invited to attend a workshop on the usage of the proposed methods, and the views are mixed. Many do not like the idea of their data being used, even after understanding that it is anonymised. Some hope the method can allow them to take tests at their local health centre instead of going to the hospital, if decisions are taken by a machine anyway, while a few others says they would stop going to controls if AI is in any way involved.

**Stage 3:** It is now time to make trade-offs, and the research group has trouble reaching consensus. Some think the potential knowledge gain outweighs the patient concerns about using anonymous data and propose they should use the data and train the model, but collaborate with some medical researchers to see if the model can identify previously unknown patterns. This would avoid using the model to make actual decisions, and thus not decrease the willingness of some patients to attend their controls. However, to meet the patients with the privacy concerns half way, they prepare an information folder on how the data anonymisation works hoping this will decrease their fears. Another group is sceptical whether any new medical knowledge would be gained and instead focuses on how the proposed model scheme could be used to improve accessibility of healthcare for those living far from the hospital. This second group additionally do not want to use the patient data, even anonymised, due to the patient concerns. Instead they want to work with the doctors at the hospital on how to encode the patterns already used into a simple and explainable model that could be used by type 2 diabetes patients to determine how often they need to attend controls at the hospital, based on self reported blood

1009 sugar measurements and medical history. In the end,
1010 the two subgroups decide to pursue their respective
1011 research direction, and both present the trade-offs
1012 they have done to the patients and doctors they
1013 talked with before, as well as in their papers.

