# OpenReview forum: "Navigating the Impossibility of Universally Ethical AI  -- a Practical Guide for Researchers"
_NLDL.org/2026/Conference — Submitted to NLDL 2026_

### Official Review · Reviewer_DWmS · 2025-10-03
**Review of Navigating the Impossibility of Universally Ethical AI -- a Practical Guide for Researchers**

**Rating:** 1
**Confidence:** 4
**Final Rating:** 2
**Final Confidence:** 4

**Summary:**

The goal of this paper is to introduce the challenges and difficulties associated with the challenge of truly using AI ethically and building AI-driven systems in an ethically conscious way.  The paper describes the authors’ argument of the moral and ethical challenges of building such a system and argues why achieving universally ethical AI is impossible.  The paper makes arguments grounded in traditional ethics theory.  The paper then provides a checklist that researchers can follow to make their everyday work more ethically conscious.

**Strengths:**

- The topic of ensuring the ethical use of AI is timely, relevant, and important.
- The author raises good points about how the broader impacts of AI research on those outside of the research community should be considered.
- The author seems to have a thorough background in ethical theory.

**Weaknesses:**

- In general, I found the prose of the paper to be difficult to follow.  The paper felt like more of an amalgamation of various topics related to ethics, applied to ML, rather than weaving a coherent story about the argument towards why the impossibility of universally ethical AI is impossible.
- Much of the content describes general ethical theory, without any connections between those topics and machine learning being justified by the author.  For example: Lines 391 to 396: “There have certainly been efforts to define a singular ethics and a universally correct way of living, but historically many such attempts have supported colonialist efforts and become an excuse to treat people with other ways of living as morally inferior.”  There should be an extended discussion of this within the context of machine learning, as that is the focus of the paper.
- Furthermore, quite a few of the arguments of the paper felt insufficiently justified.   Some examples include: For example line 370-371: “...while someone else might require 100% accuracy to deem the model trustworthy.”  These different views should be substantiated with some sources.
Another example is that it is unclear how “considering the preservation of honour” in lines 325-332 contribute to the main points of the paper: “This could be done through considering the preservation of honour, and viewing content moderation more in terms of a guardian protecting the users, rather than a form of censorship, regardless of which values are prioritised”.
- It is unclear to me what the novelty of the paper is, and what we can learn from this paper that we cannot understand from existing literature.   Much of the paper focuses on pointing out existing ethical issues in the use of AI, without providing novel insight into how ethics should be considered in the context of machine learning.  Specifically, much of the content is not novel.  This reads as more of a literature review or position paper, rather than a novel research paper.
- The core argument of the impossibility of universally ethical AI, while being the namesake of the paper, is not as thoroughly justified or argued for in the paper.
- The paper does not justify the novelty of their checklist compared to the EU assessment list or ACROCPoLis.  It is unclear why the framework proposed in the paper would lead towards a better solution to the ethical use of AI.
- Inherently, it seems to be the case that ethics is context-dependent, and that researchers must evaluate the ethics of their work carefully, including understanding the viewpoints of all of the various groups the research may impact.  Perhaps this is the “universal” truth, but I don't see how it is a novel contribution.
- The authors describe that their checklist is inspired by the EU and ACROCPoLis checklists, but do not sufficiently justify why their version of the checklist is novel, and needed.
- Given that one of the main contributions of the paper is the checklist, there should have been much more discussion on how the existing checklist (i.e., EU) are limited, and why there is a need for a new checklist.
- For the reasons stated above, I feel that the significance of the work is limited.

**Final Justification:**

While I stand by my initial points of my original justification, after reading the other reviewers' responses and the author response, I have decided to raise my score by 1. Given that it is an ethics-focused submission and may fall in the "Social and economic aspects of Machine Learning (accountability, causality, fairness, privacy, robustness, interpretability, etc.)" category, perhaps the requirements would be different from a more "traditional" ML research paper.

**Justification:**

While the authors do address an important and relevant topic (ethical usage of AI), I feel that there is not enough novel content to justify the publication of a research paper.   The majority of the paper describes existing ethical considerations in the context of AI, and broader ethical theories in general, but does not sufficiently justify why the contents of their paper are novel.  It is unclear to me what can be learned from this paper, that is not found in existing literature.  For example, it is unclear what the benefit of the introduced checklist is over existing checklists (EU).  To me, the paper reads more like a position or a literature review.   In terms of writing, I found the overall prose of the paper a bit difficult to follow.  For the aforementioned reasons, I recommend a reject decision.

---

> ### Author Rebuttal · Authors · 2025-10-17
>
> Thank you for your thorough and detailed review. I appreciate the time you took to provide such comprehensive feedback. Below, I address your main concerns point by point.
>
> 1. Purpose of the ethical background and connection to ML
> The ethical theory sections were included to illustrate the diversity and internal contradictions within ethical thinking—forming the foundation for the central argument that a universally ethical AI is impossible. I recognize, however, that this connection could be made more explicit.  I will revise these sections to better connect the ethical discussions—such as the historical attempts at universal ethics—to specific ML issues, including data governance, fairness metrics, and bias in model design. The cited sentence on colonial legacies, for instance, will be expanded to show how such assumptions can reappear in dataset construction and model deployment. The “preservation of honour” example will be clarified as an illustration of alternative value systems that contrast with commonly emphasized values such as transparency and autonomy.
>
> To further strengthen coherence, I plan to add an outline paragraph early in the paper clarifying the purpose of introducing these frameworks, and how they are used in the later arguments. In addition, I will include a taxonomy-style figure illustrating how the different ethical frameworks connect, contrast, and lead to the impossibility argument. These updates where sugested by another reviewer, and I beklieve this addition helps adress your concerns about this section as well. Together, these changes will help the reader better understand the flow of reasoning and how ethical theory underpins the ML-focused discussion.
>
>
> 2. Contribution and novelty
> The novelty of the paper lies in synthesising established ethical theories and perspectives into a coherent and accessible form for a technically oriented ML audience. Rather than proposing a new ethical framework, the contribution is to translate these theories into the context of ML research, and to demonstrate how they can be reflected upon during the research process. The goal is to bridge the gap between philosophical AI ethics and everyday ML practice by making complex ethical reasoning practically usable and relevant to individual researchers.
>
> I agree that the distinction between the proposed checklist and existing ones (e.g., the EU assessment list, ACROCPoLis) could be made clearer. These existing frameworks are extensive, often targeted at organizations and policymakers, and focus on what ethical requirements should be met. In contrast, my checklist is intended as a map indicating where in the research workflow ethical reflection can be meaningfully integrated. It is not an alternative to established guidelines, but a complement designed to make those principles actionable within individual research settings. I will expand this section to emphasize this distinction more clearly.
>
>
>
> Once again, thank you for your constructive feedback. Your comments will help me make the manuscript more focused, readable, and relevant to both ethical and technical audiences. I believe these revisions will significantly clarify the paper’s contribution and strengthen its overall impact.

---

### Official Review · Reviewer_YLGZ · 2025-10-06
**A discussion on the impossiblity of universally ethical ai with a practical guide for researchers**

**Rating:** 2
**Confidence:** 2

**Summary:**

This work is focused on the topic of ethics in AI, and argues that universally ethical AI is an impossibility. This is exemplified, discussed, and explained through a survey of established approaches to AI ethics. Afterwards, democracy-based approaches to AI ethics and two existing frameworks are presented and discussed. Based on these approaches and frameworks, it is argued that AI ethics should be focused on making conscious trade-offs and communicated them clearly. Lastly, a practical guide for researchers is presented.

**Strengths:**

1. With the recent advances in AI and its application in a wide range of different fields, it is undoubtedly important to improve our understanding of how to develop ethical AI. This article gives both a nice overview of existing approaches to the development of ethical AI, and points the way towards future approaches.
2. The article presents complex topics and challenging scenarios in an understandable and manner.
3. I found the manuscript to be clearly written, with interesting explanations and examples, and with a nice summary and checklist towards the end that could provide useful considerations for AI researchers.

**Weaknesses:**

My main concern with this work is the scope. The manuscript does not have any technical or methodological contributions, which of course is not the goal of the paper. Rather, the summary, the analysis, and the suggested future directions for developing ethical AI would have to be considered the contributions of the paper and evaluated for its originality. As someone coming from a technical background, I find it hard to judge if these contributions constitute novel contributions or mostly a summary of existing works. The first thesis of the paper is that ethical AI is an impossibility, but looking through the literature it appears that there is awareness of this impossibility, both in the mild sense [1] but also taking it further than what is written in this manuscript [2]. And in terms of the practical guidelines, there also seems to be many existing guides, for instances summarized by Corrêa et al. [3]. Therefore, the novelty of the contributions of this work is unclear.

[1] Robinson, Moral disagreement and artificial intelligence, AI & SOCIETY 2023
[2] Munn, The uselessness of AI ethics, AI and Ethics, 2022.
[3] Corrêa et al, Worldwide AI ethics: A review of 200 guidelines and recommendations for AI governance, Patterns 2023.

**Justification:**

In general, I found this paper well-written and focusing on an important topic. I think it can lead to interesting discussions at the conference. The call for papers does encourage works on "Social and economic aspects of Machine Learning (accountability, causality, fairness, privacy, robustness, interpretability, etc.)", which it is possible to argue that this paper falls under. At the same time, I find it difficult to asses if the contributions of this work are new, or if these thoughts and ideas are already somewhat well established in the community. Based on this, I am leaning towards rejecting the manuscript, but I am also open about my lack of background in the domain and willing to discuss the fittingness and novelty of the work.

---

> ### Author Rebuttal · Authors · 2025-10-17
>
> Thank you for your thorough review of the manuscript, and I am happy to hear that the writing style was appreciated. While several guides exist, I argue that their current form makes them difficult to apply within individual research contexts, particularly for ML researchers working outside of large corporate or policy settings. The practical guide is not intended as a checklist of ‘ethical compliance’, or an alternative to the established, more in-depth guidelines, but rather as a map of where in the research workflow ethical reflection can be meaningfully integrated. However, I will clarify this in the revised manuscript.
>
> The novelty of the paper is to present the complexity of ethical AI to a technically oriented ML audience, translating ethical theories into a machine learning context and providing a guide to include these considerations in practice.
>
> Finally, I appreciate your examples of other papers in the area. I have not previously read The uselesness of AI ethics, but will definitely review it, along with the other cited papers, to better position this work within the existing literature.
>
> Thank you again for your feedback, I appreciate you taking the time to carefully read my submitted manuscript, and provide actionable comments.

---

### Official Review · Reviewer_AF2x · 2025-10-07
**Paper on the different approaches that can be followed towards ethical AI**

**Rating:** 2
**Confidence:** 3
**Final Rating:** 2
**Final Confidence:** 4

**Summary:**

The paper is about different approaches and concerns that can arise when one tries to develop AI that is ethical.  In this direction the author has two main theses:

(1) It is impossible to find a universal definition for ethical AI, and

(2) We should be making conscious tradeoffs and communicate such tradeoffs to those affected by the developed AI methods.

The author discusses utilitarianism and guidelines as two different mainstream approaches for ethical AI and observes limitations of such approaches.  In sequence the author discusses other approaches for ethical AI, such as care ethics, environmental ethics, and incorporation of non-WEIRD (Western, Educated, Industrialised, Rich, Democratic) values.  After that part, the author discusses about inherent contradictions in ethical AI (e.g., contradicting definitions of fairness).  Then, it is suggested that one way of fighting against such contradictions is to follow a more political stance against these things, thus requiring democratic regulation, transparency, and participatory design.  To complement this approach the author presents two frameworks for analysing the ethical implications of AI models: (I) trustworthy AI assessment lists (e.g., distributed vie organizations/EU), and (ii) ACROCPoLis, a framework that brings together Actors, Context, Resources, Outcome, Criteria, Power, and Links between them.  Finally the author concludes with a practical guide that can be followed towards the development of ethical AI systems and help developers make informed decisions and provide the necessary tradeoffs that are being hidden in the various decisions to the people that will be affected by the developed AI system.

**Strengths:**

**S1.** I think the main strength of the paper is the practical guide that is being offered near the end of the paper.

**Weaknesses:**

**W1.** I think that this paper has an opportunity with the, usually mundane, "paper outline" to stand out.  After the introduction, it would be great if the author could provide a paragraph explaining the underlying themes of the different sections and how these are intertwined so that they can communicate the message of the author and make the flow of the whole paper much more natural. While such a paragraph with proper explanations may very well take half a column from the paper, in my opinion this is well worth it so that the meaning can be conveyed in an easier way.

**W2.** Minor typos here and there:
- "cannot" should be one word: Lines 004, 143, 380,m469,
- I think "Further" should be replaced by "Furthermore" in Lines 71, 150, 169.
- Line 114: for (t?)his monster <--- missing "t" in front of "his"?
- Line 191: have -> has
- Line 197: affect -> affects
- Line 274: serious -> seriously
- Lines 462-466: please cite the venue and do not make an exercise for the reader by googling the quote that you have in the following paragraph
- Line 486: As time goes and ... <-- missing a "by" after "goes"?
- Line 566: please fix the reference
- Line 612: this -> thus

**Final Justification:**

I appreciate the response by the author, both towards me, as well as to the rest of the reviewers.  Having said that, I believe that all the reviewers had some difficulty with the flow of the paper, which I believe is the main issue with the current paper.  So, I am raising my confidence, from 3 to 4.  I think the next version of this paper will make it an interesting paper.

**Justification:**

Perhaps I am not very used to such philosophical papers but I found it hard to follow at times.  I honestly believe that the author can improve the presentation using the "outline" paragraph I mentioned above.  Also, perhaps another important ingredient would be some figure that provides a taxonomy of some sort, visualizing what I have tried to include in my summary above.  This is clearly something that the author can do much better than me.  I just have the feeling that something is missing so that the whole paper can be appreciated better.  I believe that the author can tell a nice story, but in my opinion something needs to change in the current presentation.

Relatedly to the above comment, it is unclear to me if this is the appropriate venue for this kind of work.  Perhaps some of the journals or conferences for ethical AI that the author is mentioning in the beginning of the paper are actually more appropriate for such a submission.

---

> ### Author Rebuttal · Authors · 2025-10-17
>
> Thank you for your thorough review. I appreciate your recommendations for an outline paragraph and a taxonomy figure, I agree that this can greatly improve the paper. I will construct a figure to showcase how the different ethical frameworks connect and disagree, and add an outline clarifying the purpose of stating the different ethical frameworks and how this then is used in the later arguments. Thank you also for listing the minor typos, even after going through it several times some grammar mistakes always seem to remain.
>
>
> The main contribution of the paper is to synthesise different views on AI ethics, based on ethical theory, in a coherent way to showcase the complexity of the issues. Most of the points are definitely well-known in the community of AI ethics, but one of the issues I hoped to help remedy with this paper was the discrepancy between research on AI and research on AI ethics, where AI conferences have a lot of papers "solving explainability" or "guaranteeing privacy" based on very specific definitions of these concepts, often without acknowledging the vast amount of different views on these topics. Fostering more high-level discussions in the ML community about the goals of ethical AI, I hope to inspire other researchers to a more holistic view on their research topics. The summarising nature of the paper aims to present these issues in a way that an ML researcher can quickly grasp and apply to their research, with the necessary ethical theory included to make the paper and guide self-contained. While researchers could study these topics in depth, the paper aims to provide an accessible synthesis for those who may not have the time to do so. Thus, this paper contributes by bridging philosophical debates and ML research, and by summarising these insights into actionable steps to inform the research process.
>
>  I acknowledge that this is not the typical, technical NLDL contribution, but I believe it aligns with the NLDL interest in ethical and societal dimensions of deep learning.
>
> Thank you again for your feedback, I will incorporate it to improve the paper's clarity.

---

### Meta-Review · Area_Chair_GjGY · 2025-10-30

**Recommendation:** Reject
**Confidence:** 4

**Metareview:**

The reviewers agree that the paper tackles an important and timely topic for the machine learning community. Its exploration of ethical frameworks, the argument against a "universally ethical AI," and the inclusion of a practical guide were recognized as valuable elements.

However, Reviewers (AF2x, DWmS) found it challenging to follow (loosely connected topics rather than a coherent argument).
Furthermore, Reviewers (YLGZ, DWmS) raised concerns about the paper's novelty. It was unclear how the paper's core arguments or its proposed checklist represented a novel contribution beyond existing literature and established guidelines, such as the EU's assessment list.

Although the author's rebuttal proposed helpful revisions, the paper must be judged as submitted. In its current form, its weak narrative structure and insufficiently articulated novelty limit its overall contribution.

---

### Decision · Program_Chairs · 2025-11-05

**Decision:**

Reject

**Comment:**

Based on the reviewers and AC comments, the paper cannot be presented at the conference.